# Frontier Markets and Sustainable Entrepreneurial Competences: An Exploratory Study of the Impact of a New Industry in Guatemala

Jose Godinez * and Denise R. Dunlap *

Manning School of Business, University of Massachusetts Lowell, One University Avenue,
Lowell, MA 01854, USA
* Correspondence: Jose_Godinez@uml.edu (J.G.); Denise_Dunlap@uml.edu (D.R.D.)

**Abstract:** There is growing interest among scholars and policy makers to develop sustainable entrepreneurial competences in pre-emerging, frontier markets characterized by limited access to advanced capital, high protectionism, and weak formal institutional environments. To become internationally competitive, these markets need to radically rethink their long-standing, embedded practices, which have often been linked to socioeconomic inequality. Our study, grounded in corporate entrepreneurship, is an exploratory analysis of why and how well-established firms, operating in the financial service industry, created more equity-based businesses practices to enter the new industry of mobile banking. The firms in our study needed a combination of both economic incentives and social pressures to do so but, in the process, developed new entrepreneurial competencies. Successful firms were those that significantly altered their embedded practices and engaged in fostering new informal relationships with previously overlooked stakeholders, particularly customers from indigenous backgrounds. Our multi-case, inductive research design offers theoretical and practical insights regarding how incorporating internal and external corporate entrepreneurial factors in an underserved market setting, such as the frontier market of Guatemala, not only fosters socioeconomic equality but also creates international attractiveness and competitiveness.

**Keywords:** corporate entrepreneurship; banking; sustainable development goals; emerging markets; business ethics; socioeconomic inequalities; international competitiveness



## 1. Introduction

*"In Guatemala, access to finance has greatly expanded over the past few years"* [1].

The number of financially excluded people around the world remains high and totaled about 1.7 billion in 2020 [2] despite many efforts by governments, international agencies, and private businesses to promote financial inclusion. Financially excluded populations concentrate mainly in less advanced countries and lack access to financial services, such as bank accounts, credit, savings, loans, and insurance coverage [3,4]. The consequences of financial exclusion in each location are an increased gap in wealth inequality and slow economic growth [5]. In addition, those without access to formal financial services may resort to dealing with informal, predatory lenders that charge exorbitant interest rates [6].

To this end, the United Nations (UN) created the Sustainable Development Goals (SDGs) to improve the living conditions of its inhabitants and to protect the environment [7]. Three of these goals (i.e., SDG1 No Poverty; SDG 8 Decent Work and Economic Growth; and SDG10 Reduced Inequality) are akin to ending financial exclusion [4]. Specifically, the UN seeks to *"support banks, cooperatives, microfinance institutions, money transfer companies, and mobile network operators to extend the reach of financial markets where they would otherwise not go"* [2]. Despite consensus about the serious challenges resulting from rising global inequality, research that examines the mechanisms behind organizational decision making in less advanced countries is still complex. Scholarly inquiry also remains in its infancy

since organizational embedded practices continue to be largely responsible for denying a large and financially oppressed population the opportunity to advance their socioeconomic status [8].

Large, well-established firms, in this context, are often the slowest to adopt change and have the most difficulty when it comes to radically reformulating their highly embedded practices to improve greater representation of disadvantages groups (e.g., [8]). A more nuanced understanding of organizational embedded practices is important [9] since they model the world, rules, norms, and procedures that are at least partially known by all members of an organization [10]. While creating substantive changes to existing business practices that embrace greater diversity and inclusion is a global objective, it is less common among firms operating in less advanced economies and even rarer in the context of frontier markets [8]. Yet, if companies want to become more globally competitive, establishing practices of greater inclusion is necessary.

In this study, we adopt a corporate entrepreneurship (CE) lens to analyze the important role that entrepreneurship plays in disrupting outdated embedded practices [11]. Corporate entrepreneurship is defined as the efforts of established firms to create and enact new ideas, growth prospects, or business models by recognizing and pursuing opportunities to improve their operations and/or by commercializing new products in new markets [12]. Scholars have placed special interest on corporate entrepreneurship research because of its importance for strategic renewal [13] and the development of new ethical competencies that have the potential to upgrade and revise a firm's competitiveness while at the same time sustain its advantages [14]. An underdeveloped yet fruitful area of corporate entrepreneurship scholarship has yet to systematically disentangle the different types of CE, internal and external, [12] and how they can in combination work together to support managers operating in frontier markets, characterized by an abundance of sheltered industries that seek to overcome decades of organizational resistance to changing the status quo [15].

Advancing this research within the context of frontier markets is of growing importance since such knowledge could help policy makers increase the attractiveness of these countries among international investors. These pre-emerging markets not only face significant socioeconomic inequalities, but they are smaller, riskier, and more resource-constrained, which hinders their economic development. They also face weaker formal institutions and have been considerably less studied in the literature compared to the larger and more attractive emerging markets, known as the BRICs (i.e., Brazil, Russia, India, China) [16]. While frontier markets are more developed than the least-developed, they are still significantly underdeveloped [17] and attract less foreign direct investment due to their strong protectionism over certain sheltered and protected industries of national interest [18].

Although frontier markets have received less attention in the literature, they offer tremendous opportunities for entrepreneurial firms to enact inclusion [19] since they account for a quarter of the global land area, approximately 20% of the population, and $100 billion annually in purchasing power [20]. Frontier markets are also changing and becoming more open to global competition [21]. Therefore, to remain internationally competitive and to further advance their pre-emerging status, even the most well-established firms operating within this context need to change their embedded practices to create greater inclusiveness. In this vein, a better understanding of what kinds of entrepreneurial competencies and skills are necessary to enact change is imperative especially since these markets are fraught with uncertainty and risk, and the firms operating in these locations have limited resources to do so [22].

To conduct our study, we analyzed all firms operating within the banking industry in Guatemala that had well-established embedded practices but nevertheless utilized corporate entrepreneurship to change those practices to enter the new mobile banking (M-banking) industry to reach traditionally excluded populations. Guatemala is a frontier market characterized by firms with embedded exclusionary and discriminatory business

practices that have been present in the country for over four hundred years [23]. Within the last decade, as new industries have begun to emerge, a mere handful of established firms have started to realize the dangers of holding onto these archaic embedded practices [24] and thus have begun utilizing corporate entrepreneurship to change them. To explore this evolving phenomenon, the purpose of our study was to analyze how and why firms changed three embedded practices of exclusion characteristic of firms operating in Guatemala: exclusionary and discriminatory practices with potential clients, particularly those from indigenous descent; vertical integration to maintain control of the value chain through supplier relationships; and reliance on informal relations with members of the government elite to protect their operations.

Our study makes several contributions to literature. First, we examine a critical business ethical issue that is driving inequalities around the world. We study organizations practices that have normalized and perpetuated the exclusion and discrimination of oppressed groups. We examine how corporate entrepreneurship can act as an important enabler of socioeconomic upliftment and change. We conceptualize those at the forefront of change as a process whereby firms operating in relative stability utilized corporate entrepreneurship to reform their embedded practices to enter a nascent industry in a frontier market. Our results show that financial incentives alone were not responsible for their desire to develop inclusionary practices in part because they were financially stable before they decided to enter the newly established M-banking industry in the country. Change, however, did occur and was the result of a combination of economic incentives and external social pressures. Second, we found counter-intuitive evidence that intra-firm cooperation [25] and the cultivation of once traditionally beneficial linkages with high-ranking members of the local government elite [26] were not efficient or productive. The firms we analyzed took on a radically different and entrepreneurial approach by leveraging informal linkages with previously overlooked local actors: technological suppliers, users with little formal education, and low-level government bureaucrats. Third, we describe the underlying process whereby well-established firms developed new entrepreneurial competencies. By radically changing their embedded practices to serve the needs of the traditionally and financially under-represented population groups in a less advanced economy, these firms played a critical role in the reduction of socioeconomic disparities, which is one of the key goals of the UN's Sustainable Development Goals.

The rest of our paper is organized as follows. In the "Theoretical Background" section, we present the literature that contributed to our understanding of corporate entrepreneurship in frontier markets. We then describe our methodological approach, our sample, and setting in the "Methods" section, followed by a description of our findings in the "Results" section. In the "Discussion" and "Conclusions" sections, we summarize our contributions and discuss the study's possible limitations and highlight future avenues for research.

## 2. Theoretical Background

### 2.1. Corporate Entrepreneurship: Internal vs. External

Corporate entrepreneurship pertains to the efforts of established firms to conceive and pursue new ideas, business models, and expansion prospects by uncovering and following opportunities to upgrade operations, create and market new products, and enter new geographic areas [12]. We utilized a corporate entrepreneurship framework to conduct our study because it allows us the opportunity to understand the processes whereby organizations adopt entrepreneurial strategies to change embedded practices [27] in a frontier market setting and, by doing so, lead to greater financial inclusion among stakeholders within the country. In this context, changing embedded practices in a frontier market means developing innovative capabilities to explore and exploit ever-expanding opportunities [28] that can become standard within an organization [10] while, at the same time, operating in a market characterized by static business practices, discrimination, financial exclusion, protectionism, and a weak formal institutional environment [29].

There are two broad kinds of corporate entrepreneurship, internal and external [30]. Internal CE focuses on how the firm utilizes resources and capabilities to foster innovation, improve operations, capitalize opportunities, and create new services and products [31]. Often, internal CE is characterized by risk taking, innovativeness, and proactiveness [32]. External CE focuses on extending the boundaries of the firm [33] mainly by acquiring, cooperating with, or establishing new subsidiaries [34].

External and internal corporate entrepreneurship are not entirely opposed to one another [35]. Firms can pursue external and internal CE concurrently since these activities can reinforce and complement each other [36]. In this regard, firms can pursue external and internal CE activities at unison by utilizing different resources and skills. Nevertheless, internal and external CE are risky and potentially expensive [12]. For example, internal CE might require expanding and reorganizing the firm's human capital and the adoption of new management practices, while external CE may include establishing new subsidiaries and venturing into new geographic areas [37]. Due to the risk and expense that CE, internal and external, represents to firms, those organizations operating in sheltered industries characteristic of some frontier markets [38] may not have incentives to change their embedded practices, especially if these firms see themselves as already well established and successful.

## 2.2. Corporate Entrepreneurship in Frontier Markets

Extant scholarship analyzing corporate entrepreneurship proposes that organizations initiate changes to their practices to capitalize on new entrepreneurial opportunities [27] and remain competitive in highly contested markets [39]. However, current literature has not yet uncovered why and how established firms in less advanced economies choose to proactively engage in changing their age-old embedded practices, especially since maintaining the status quo, in most cases, has led to their success [40]. Achieving such success, however, comes at a societal cost and has led to wide-spread discrimination and socioeconomic inequalities [1]. When firms operate in locations characterized by protectionism and weak formal institutional environments, there are little incentives to change their embedded practices. Yet, the opportunity to exploit entrepreneurial opportunities in new technology sectors can be a motivating force behind why some established firms are slowly rethinking and changing their embedded practices [41], which in turn have the potential to bring about greater financial inclusion and also address century-long socioeconomic inequalities.

In order to enact changes to already existing and firmly embedded practices within well-established firms, particularly in less advanced markets, such as frontier markets, firms may need to acquire and configure new entrepreneurial competencies and skills compared to those operating in developed countries. Moreover, the sources of resource acquisition and associated resource configurational processes for such firms may differ considerably [42] based on both internal and external corporate entrepreneurship approaches and resource constraints. Internally, it may be the case that frontier-market firms prefer to reconfigure existing resources instead of engaging in knowledge-creation processes [37]. Externally speaking, these firms may choose to leverage their existing networks as opposed to acquiring or establishing new subsidiaries [42]. Building on these arguments, we focus on how frontier-market firms configure and develop new entrepreneurial competencies so as to cooperate better with key external stakeholders, like customers and suppliers. We study the complexities of developing these competences since they are among the least understood aspects of corporate entrepreneurship [43], especially in less advanced economies, such as frontier markets compared to emerging markets.

Another important distinguishing factor about frontier markets compared to more advanced economies that is worth noting is that they are characterized by financial exclusion, protectionism, and a weak formal institutional environment, all of which may not be properly enforced by current laws and regulations [44]. As such and within these protected environments, established firms have a reputation of advancing their competitive advantage by leveraging their power and influence over high-level public officials that are willing to give them special treatment and provide them with valuable resources [45]. To

this end and building on the extant literature, there remains insufficient knowledge about corporate entrepreneurship in well-established firms operating in frontier markets and what motivates them to utilize entrepreneurial practices (both internal and external) to make radical changes to previously held, embedded practices that once enhanced their revenues streams by explicitly perpetuating inequalities. In the pursuit of extending scholarship in this important context, we engage in a multi-case inductive research design to uncover the mechanisms behind what influences established firms to selectively move away from discrimination-based practices towards more inclusionary programs especially when entering a new industry.

## 3. Materials and Methods

### 3.1. Research Design

We utilized a qualitative approach to analyze all banking firms in Guatemala. These firms were generating profits at the time the research was conducted and continued to enter the new M-banking industry. They were in full operation in the country at the time when the research was conducted. Firms that exited the market, previous to this research, were not included in the sample. We followed Eisenhardt [46], who called for examining a new phenomenon qualitatively at the country level. This exploratory approach allowed us to understand and explain variations in the patterns and pace of these firms' corporate entrepreneurial activities compared with others in the literature [47]. Limited research has also been conducted on frontier markets in this sector in Latin America, although the region has seen an expansion on entrepreneurship research, and none in Guatemala, where our research took place.

### 3.2. Research Context

We defined our field geographically [48] and chose Guatemala as our setting because entrepreneurship rarely fits all regional contexts, and relevant analyses should be performed at the country level [49]. Guatemala is a frontier market characterized by financial exclusion, an unstable formal institutional environment with an attractive but unregulated nascent M-banking industry. Significant progress was made in this regard, especially from 2011 to 2017 (i.e., the number of people with bank accounts nearly doubled in Guatemala). Yet, in 2019, only 44% of the population currently had a bank account [1]. The establishment of this industry in Guatemala can be traced back to the large number of remittances that the country receives from expatriates. These expatriates are responsible for sending nearly $10 billion per year in electronic transactions, on average, which is approximately 10% of the country's GDP [50]. Additionally, in 2018, there were about 1.47 million consumer accounts affiliated with M-banking, which consisted of 12,435 mobile transactions worth $1.11 million in total. These transactions constituted an average profitability rate of 12% for the suppliers of these services [51]. In addition, it is important to note that telecom penetration was well developed, with about 1.5 mobile telephones per person [52]. However, despite the potential opportunity that these factors suggest, Guatemala is a frontier market characterized by high protectionism, toward large-established firms [23], and an unstable institutional environment [29]. In fact, in 2015, the Guatemalan president and vice-president were forced to step down and were further incarcerated over their law-breaking practices [53]. As a consequence, a transitional government was formed [54], and elections were held in 2016. In 2017, a new government was established, but its mandate was deemed illegitimate due to allegations of corruption [55].

### 3.3. Examination of Embedded Practices

Traditionally, firms operating in Guatemala have not been very entrepreneurial [23]. This is due in part to the fact that Guatemala's economy has been highly sheltered and closed to foreign competitors [56]. This unique setting helped firms foster three specific embedded practices: (a) exclusionary, discriminatory practices with potential clients, particularly those from indigenous descent; (b) vertical integration to maintain control of the

value chain; and (c) strong informal relations with members of the government elite to protect operations [57]. Given their complex nature, we examined whether and to what extent firms entering this new industry utilized corporate entrepreneurial strategies. Further, we explored how the process of doing so led to the development new competencies and the change of outdated, embedded practices.

### 3.4. Data Collection and Design

In 2015, two years before data collection began, we rigorously desk-researched and explored the established firms that began operations in the M-banking industry in the country. We focused on their public annual reports from 2006 to 2017, which is the time this industry was first introduced to Guatemala, until we conducted our fieldwork. We also reviewed industry publications as well as official government documents. These documents allowed us to corroborate public statements. They also helped us to validate the chronology of events, provide details not expressed during the interviews, and offer a corroboration method of debates and discussions. This procedure allowed us to confirm the validity of private and public sources due to their similarity. Further, it is worth noting that during this time period, there were only six established firms that had long-established operations in the country and began operating in this industry in 2006. We included all of the firms that had operations at this specific time in our inquiry. In total, we conducted 37 personal interviews. Twenty-one interviews were conducted with all the firms operating in this industry in Guatemala in 2017, and the remainder, 16, were interviews granted by members of the local government. Table 1 presents the backgrounds of the firms analyzed, and Table 2 presents the profiles of the public servants interviewed.

**Table 1.** Profile of respondents (firms).

| Respondent | Firm | Year/Place Founded | Year Established in Guatemala | Year Company Established M-Banking | Respondent's Role |
|---|---|---|---|---|---|
| 1 | A | 1971 Guatemala | 1971 | 2008 | Guatemalan Director |
| 2 | A | | | | SM Strategy |
| 3 | A | | | | MD Operations |
| 4 | B | 1962 Guatemala | 1962 | 2006 | MD Operations |
| 5 | B | | | | Guatemalan Director |
| 6 | B | | | | MD Operations |
| 7 | B | | | | SM Strategy |
| 8 | C | 1966 Guatemala | 1966 | 2006 | Guatemalan Director |
| 9 | C | | | | MD Operations |
| 10 | C | | | | MD Operations |
| 11 | D | 1967 Guatemala | 1967 | 2008 | Guatemalan Director |
| 12 | D | | | | SM Strategy |
| 13 | D | | | | MD Operations |
| 14 | D | | | | MD Operations |
| 15 | E | 1952 Nicaragua | 1990 | 2007 | Guatemalan Director |
| 16 | E | | | | MD Operations |

**Table 1.** *Cont.*

| Respondent | Firm | Year/Place Founded | Year Established in Guatemala | Year Company Established M-Banking | Respondent's Role |
|---|---|---|---|---|---|
| 17 | E | | | | SM Strategy |
| 18 | E | | | | SM Strategy |
| 19 | F | 1990 Sweden | 2004 | 2006 | Guatemalan Director |
| 20 | F | | | | MD Operations |
| 21 | F | | | | SM Strategy |

Note: SM indicates Senior Manager, and MD indicates Managing Director.

**Table 2.** Profile of respondents (members of local government).

| Respondent | Institution | Respondent's Role |
|---|---|---|
| 22 | Congress | President of Commission of Telecommunications, Transportation, and Public Works |
| 23 | Congress | Vice-President of Commission of Telecommunications, Transportation, and Public Works |
| 24 | Congress | Secretary of Commission of Telecommunications, Transportation, and Public Works |
| 25 | Congress | Member of Commission of Telecommunications, Transportation, and Public Works |
| 26 | Ministry of Comm., Infrastructure, and Housing | Upper-level public official |
| 27 | Ministry of Comm., Infrastructure, and Housing | Upper-level public official |
| 28 | Bank Superintendence | Upper-level public official |
| 29 | Bank Superintendence | Upper-level public official |
| 30 | Supreme Court Justice | Member, Civil Court Council |
| 31 | Supreme Court Justice | Member, Civil Court Council |
| 32 | Supreme Court Justice | Magistrate |
| 33 | Supreme Court Justice | Magistrate |
| 34 | Ministry of Comm., Infrastructure, and Housing | Bureaucrat |
| 35 | Ministry of Comm., Infrastructure, and Housing | Bureaucrat |
| 36 | Ministry of Comm., Infrastructure, and Housing | Bureaucrat |
| 37 | Ministry of Comm., Infrastructure, and Housing | Bureaucrat |

During the summer of 2017, the lead researcher contacted all of the firms operating in Guatemala directly with the additional help of personal contacts in each firm. After a month of sending written requests, interviews were granted. The interviews took place that summer and were conducted face-to-face by the lead author. The respondents were both senior managers who worked directly on M-banking activities in each of these firms. These

interviews were designed following the methodology utilized by Lee, McGoldrick, Keeling, and Doherty [58], which proposes the use of a broad framework to understand a nascent industry rather than a single method. All of the interviews with the firms' respondents were semi-structured with open-ended questions. The questions focused on why and how these firms decided to change their embedded exclusionary practices to enter a nascent industry in an unstable institutional environment characteristic of a developing country. When the interviews took place, their semi-structured nature allowed for an increase in focus on the firms' operational strategies after questions about the general environment of the industry were first discussed. This was possible because all respondents were experts in this subject.

The interviews became more focused and provided an in-depth account of the firms' operations because of the repetitive and accumulative nature of the data-gathering process [59]. Each interview lasted between 60 and 90 min and was recorded at the offices of the respondents. All the interviews were carried out in Spanish. Relevant quotes were later translated into English and were analyzed by an external researcher to verify accuracy. The interviews were conducted following a standard protocol to obtain emerging themes [60] and were transcribed verbatim.

Although the M-banking industry in Guatemala is not directly regulated, all firms in the country are subject to the local laws. Hence, to understand the issue from the regulators' perspective, interviews were conducted with key members of the government, namely members of the government elite, judiciary, and bureaucracy. Scholars have regarded governments as either an obstacle or an aid to entrepreneurship and innovation [61]. For this reason, Mauro [61] argued that members of the government elite (i.e., those of the upper echelon of government) in a location play a crucial role in maintaining economic stability depending on whether they aid or block entrepreneurial firms. In addition, Mauro [61] argued that lower-level bureaucrats play an important role in aiding entrepreneurial firms since they can expedite or delay the approvals and permits needed to operate. Finally, North [62] emphasized the value of an effective judiciary system to boost economic performance by enforcing contracts. Thus, to understand their influence in the operations of the firms entering the M-banking system, we interviewed members of the three categories of government officials.

Government officials interviewed included four legislators working in the Commission of Telecommunications, Transportation, and Public Works of the Guatemalan Congress and four members of the government elite: two in the Ministry of Communications, Infrastructure, and Housing and two in the Bank Superintendence. We also conducted four interviews with members of the Legislative Organization and four interviews with bureaucrats of the Ministry of Communications, Infrastructure, and Housing. These informants were selected due to their role in the industry being studied. The interviews were conducted in the summer of 2017, and each interview lasted between 40 and 90 min, abided by a standard protocol to obtain emerging themes in field research [60]. Interviews were conducted in Spanish. In this phase of the data collection, interviewees did not allow for the interviews to be recorded. Instead, notes were taken and then reviewed by the respondents. The illustrative quotes from those notes were later translated into English and were also analyzed by an external researcher to ensure accuracy.

### 3.5. Data Reduction

We approached our analysis with a multi-case inductive research design, which scholars recommend as appropriate for theoretical development [63]. We chose this method because it allows for replication. In other words, these cases serve as a series of experiments that can prove or disprove arising conceptual observations [64]. A key factor of multiple-case design is that it allows for better generalization than single-case studies. Hence, theoretical insights derived from this research design can be validated and extended in future research with the help of alternative methods [65]. In this phase of the research, we began by ensuring that all of the sampled firms had the three embedded practices

described before. We then analyzed why and how these firms utilized internal and external corporate entrepreneurship to change those practices to enter and operate in this nascent banking industry in Guatemala.

*3.6. Coding Methods*

We began coding interview transcripts following recommended practices for qualitative methods [66] with the help of Nvivo. We began with open coding to break data down in discrete parts to be examined and compared for similarities and differences [60]. In total, we found that all of the sampled firms had embedded practices of exclusion to a certain degree with possible clients, some aspect of vertical integration, and relations with members of the government elite in Guatemala. During this phase, we coded the description of the practices and linked those fragments from related categories. We identified patterns in the respondent's narratives regarding "usual" practices in their operations before entering the M-banking industry. For instance, we labeled statements reflecting disdain for possible clients of indigenous descent as "exclusionary practices." Practices of internalizing activities were labeled "vertical integration." Finally, establishing informal relations with members of the government elite were labeled "relations with elite."

*3.7. Analysis of Data*

During the next phase, we analyzed the data for motives and actions involving some type of change. We followed Lok [67], who argued that theory-developing methodology should not focus on obvious, explicit self-identifications since previous research has demonstrated it to be particularly important. Instead, we focused on the skills and mechanisms that allowed established firms to initiate change through corporate entrepreneurship [68]. Hence, we incorporated different levels of external pressures to analyze the why question and different levels of changes to embedded practices to analyze the how. We did this by line-by-line coding, followed by axial coding to increase the depth in categories [69]. We applied open coding to identify themes and patterns that would then be used for cross-case comparison [60]. We adopted an abductive approach, as proposed by Dubois and Gadde [70] for case research, and initial codes were informed from existing theory. Those codes included important keywords, such as "motives," "incentives," and "restrictions." Subsequent codes arose from the data and other relevant theories. These codes included important keywords, such as "economic potential," "society pressures," "laws," "regulations," "creating knowledge," and "generating stability." The coding was, hence, partly adaptive to reflect ideas that emerged inductively from the interviews themselves [71].

Once our coding was finalized, the lead researcher interpreted the data using previous knowledge of the Guatemalan context and the data gathered [72]. As a result of several iterations between the data, summaries, and theory, we were able to generate two explanations or dynamics that explained why established firms operating in this frontier-market country decided to change their embedded practices. We labeled these as economic incentives and social pressures. To answer the how question (i.e., how established firms changed their embedded practices), we generated two additional dynamics, which we labeled as new sources of knowledge and new sources of stability.

## 4. Results

We initially focused our analysis on the circumstances that prompted established firms to seek change in their embedded practices in a manner that reduced financial exclusion. Here, we present why established firms entered this nascent industry in Guatemala. Our analysis proposes that their motives were a combination of economic incentives and social pressures. We then present how established firms were able to utilize internal and external corporate entrepreneurial strategies to operate in a new industry in an unregulated, developing country. Our findings show that to succeed in a new industry in an uncertain institutional environment, established firms leveraged new corporate entrepreneurial

competencies to uncover new sources of knowledge (internal CE) and new sources of stability (external CE) without increasing their operational costs.

### 4.1. Reasons for Changing Embedded Practices

Extant theory has argued that corporate entrepreneurship in established firms with embedded practices is rare since actors who have conformed to a set of established practices might not be able to envision new ones [73]. In this context, the more the embedded practices are firmly rooted in a firm, the more resistant a firm is to change them [74]. Additionally, if this firm also operates in a frontier market, characterized by exclusion and high protectionism, its incentives to change them diminishes even further. Interestingly, our results show some counter-intuitive findings. We found that while entering a new industry in a less advanced economy was complex, it still offered new entrepreneurial opportunities to reduce systemic inequalities. In this context, however, firms required a combination of external incentives and social pressures to refocus their embedded practices to be more inclusionary.

### 4.2. Economic Incentives

Firms can adapt their approaches to take advantage of economic opportunities [75]. Nevertheless, less is known about why a firm with economic stability would want to develop new entrepreneurial competencies to enter in a new industry, which could radically change the nature of its existing embedded practices. Our results show that the firms analyzed were generating profits before entering this industry. These profits were generated mainly due to the protectionism characteristic of the Guatemalan government towards large firms, which discourages competition in certain key industries, such as banking and communications [76].

In spite of such nationalistic measures, Guatemala is mainly a cash-based society [23], and as such, banks have had "*a hard time expanding operations within the country,*" according to Respondent 15. However, once remittances from expatriates began increasing in the country, established firms in the banking and mobile phone industries could not ignore them. For banking firms, remittances were a perfect opportunity to expand operations to traditionally unbaked populations. For the mobile phone company in our study, facilitating remittance transactions meant being able to sell more products with a higher profit margin to their existing consumers.

To illustrate the financial opportunities posed by remittances, Respondent 11 argued that:

> "*When you see the amount of money that our compatriots are sending every year to their families, you cannot just be idle. You must see how you can take part of the action. It is not only the transfer fee but also gaining new clients, the recipients of the money now need a bank that help them save their money; a bank that gives them loans.*"

The respondents, however, knew that although the volume of remittances was high, generating a profit with their existing model that consisted of "*large stores with high overhead costs*" was not possible because the "*small profit margins could not sustain such expenses*" (Respondent 3) and the limited resources these firms had. Once mobile telephones were widely utilized in the country, the attractiveness of entering the M-banking industry increased because of the potential profits were perceived as high, and the investment to enter the new industry was perceived as low. This was expressed also by Respondent 1, who argued that "*When we saw that every Guatemalan had a cellphone in their hands, we knew we could enter the mobile banking business . . . Now our costs [to enter the new industry] were minimal, and the potential to make money from electronic transfers was great.*"

### 4.3. Social Pressures

Our results also indicated that while economic incentives were a driving force for established firms to enter this new industry, these companies needed an extra push to fully embrace it because they were already profitable in their operations. According to

all respondents, the international community with presence within Guatemala has been working in the country to end financial exclusion and financial discrimination. Foreign governments and international non-profits were engaged in an aggressive campaign that included providing tools to businesses to find cost-effective solutions to financial exclusion [77]. These entities also provided technical assistance for firms to implement the United Nations Sustainable Development Goals, which includes sustainability, equality, end of poverty, and economic growth [4]. Hence, respondents argued that the M-banking industry allowed them to help with the financial inclusion goals of the international community by providing financial services to traditionally excluded and even discriminated populations in Guatemala while at the same time also generating profits.

To illustrate this phenomenon, Respondent 9 argued that:

*"The American Embassy offered us assistance to take advantage of the [M-banking] industry to help with the exclusion problems in the country. This was a great opportunity for us to prove that we care about inclusion of the poor in the financial system . . . It is also a win-win because the American government in turn helps us promote our services to all our compatriots working in the U.S. who need to send money to their families."*

An interesting finding of our research was when respondents agreed that neither economic incentives nor social pressures on their own were enough reasons to change their embedded practices. Instead, as Respondent 19 stated, "*Once we saw the possible financial gains of this industry and the societal pressures to be more inclusive, we had no other choice than to enter the M-banking business.*"

Table 3 presents a summary of the multiple sources of data and illustrations of our findings of why established firms with embedded practices decided to enter the nascent and unregulated M-banking industry in a developing country.

**Table 3.** Data sources with illustrations of why firms changed their embedded practices.

| Dynamics | Sources | Illustrations |
|---|---|---|
| Economic incentives | Interviews with M-Banking firms<br>Press article (Villalobos-Viato and Gamarro, 2017) | The M-banking industry is the future. We can generate attractive profits if we play our cards right. (Respondent 14)<br>Mobile transactions are 13% of total transactions in the country [Guatemala]. This means there is room for growth. |
| Social Pressures | Interviews with M-Banking firms<br>World Bank (2017)<br>International Monetary Fund (2016) | The business-as-usual method is not viable anymore. Several sectors are pressuring us to be ethical and help end financial exclusion . . . the M-banking business is how we can answer to those pressures. (Respondent 1)<br>Financial inclusion is critical in reducing poverty and achieving inclusive economic growth in Guatemala . . . M-banking can be an effective tool to generate financial inclusion. |

*4.4. Implementation of Embedded Practice Changes*

4.4.1. New Sources of Knowledge

Technological Suppliers

Technology is costly and one of the greatest concerns among M-banking firms [78]. Our results show that all participating firms utilized internal CE by investing in next-generation technologies to enter the emerging industry, which represented a substantial

cost investment by the firms. Concurrently, the sampled firms utilized external CE and leverage informal relations with technological suppliers to keep abreast of their competitors' activities. Doing so yielded no additional costs to these firms. In contrast with recent literature that proposes that industry uncertainty leads to firms relying on their own R&D [79] and on other firms at the same level of the value chain [80], we found that the sampled firms invested in top technology and also leveraged their informal relations with technological suppliers to gather knowledge to enter and succeed in the industry while maintaining costs as low as possible.

We also found that cooperation was virtually non-existent among the firms analyzed due to suspicions over intellectual property theft and because of the fear of helping a competing firm. However, by establishing informal relations with technological suppliers, all respondents kept abreast of their competitors' actions, as expressed by Respondent 7:

> "*Of course, we are aware of what the competition is doing, as it is part of conducting businesses [ . . . ]. Obviously, we only know what the competition makes publicly available, but unfortunately, that is not enough . . . To solve this issue, we create clear communication channels with important suppliers. You know, there are not too many companies that supply the technology needed to operate in this industry. This means that we use the same suppliers as our competition. If we treat [our suppliers] well, and they trust us, we are likely to gain information about what our competitors are doing or will do.*"

Users with Little Formal Education

Another important step taken by the sampled firms was to utilize internal CE to reconfigure their products and external CE to leverage their relations with previously overlooked customers to gain knowledge. Striving to provide good service to customers is highly unusual in Guatemala, especially if the customers are poorly educated and/or of indigenous descent [57]. Nevertheless, the large portion of the unbanked population in the country have little education and belong to one of the several indigenous ethnicities. Thus, firms deciding to enter the emerging industry needed to change their approach and have direct relations with this previously ignored group of the population if they aimed to capture them as customers. Interestingly, all respondents agreed that they reconfigured their relations with potential customers with little formal education to receive feedback of how to better provide their new services from this important potential client group. While users could not provide technical assistance of how to develop better products and services, firms utilized external CE to gather information from these users about what they liked or did not like about particular products and services.

In a surprisingly clear departure from traditional practices in the country, firms utilized internal CE to change their products and began providing their M-banking products in the users' mother tongue. In this regard, Respondent 3 said:

> "*We cannot rely on users to help us with the technical aspects of our operations, but I do not think any firm in any industry does that. Instead, our users test our products to see if they do what they are supposed to do and if they do it easily [ . . . ]. Some people might think that the education level of our clients is a problem for us, but in fact, it is the opposite. We measure our success by making a mobile application for someone with little schooling, and now, because of their suggestions, we offer them in their native language! Of course, we cannot do that without them testing our products and services with our target market.*"

4.4.2. New Sources of Stability

To operate in a nascent industry in a frontier-market country, established firms entering it needed to utilize external CE to find new sources of stability. In this case, previously overlooked government actors provided such stability. Current scholarship argues that when firms encounter a weak formal institutional environment, one of the most effective sources to obtain stability is to establish relations with politicians to receive their protec-

tion [81]. Our results suggest that one of the three distinct groups of local government actors was the most resourceful and helpful to firms navigating in this emerging industry: members of the government elite, of the judiciary, and of the bureaucracy. We found that developing relationships with low-level bureaucrats in the country was unexpectedly the most effective way to generate the stability needed for these firms to establish successful operations.

Bureaucrats

Firms in our study expressed that they had utilized external CE to generate the stability needed to operate in an unregulated new industry in a frontier market by creating informal relations with low-level bureaucrats. Those benefits arose from the power that bureaucrats have in expediting the necessary conditions that firms needed in order to operate in Guatemala. As Respondent 6 put it, "*bureaucrats are the first point of contact when dealing with governmental institutions.*" In this study, their importance in creating the certainty needed to offset institutional uncertainty appeared to be critical. Additionally, all respondents agreed that although bureaucrats do not have policy-making responsibilities, they acted as intermediaries between members of the government elite and of the judiciary, which had the power to make the changes needed but were too unstable to be reliable.

Bureaucrats were in key positions to intervene on the behalf of firms with those with the power to do so. This was expressed by Respondent 4:

> "*The political situation in Guatemala is quite unstable. Two former presidents and a vice-president are in jail along with several ministers. The situation is a mess, but we still need to do our job [ . . . ]. Bureaucrats, however, are a source of certainty in this unstable situation. They are reliable and stay in their positions long after their bosses are gone. They act as our intermediaries between us and those in power. They inform of our needs to whoever happens to be in charge [ . . . ]. Keep the bureaucrats happy, and they will take care of you.*"

Members of the bureaucracy in Guatemala also agreed on the nature of this influence over businesses, particularly those operating in an unregulated industry. According to the bureaucracy respondents, since the formal institutional environment regulating M-banking activities was weak, businesses have no way of knowing what they could or could not do, so they spent a considerable amount of time talking to members of the bureaucracy in an effort to understand the requirements needed to operate within the country. Such requirements were found to include such items as applications for licenses and permits and tax obligations.

These interactions also appeared to have allowed the firms and bureaucrats to form bonds of trust over time, which could be exploited later by those firms. Respondent 36 provided an illustration of this process:

> "*[ . . . ] yes, I have spent quite some time talking to higher ups in this industry [M-banking]. Of course, after a while, you get to know them, and they get to know you. This relationship is beneficial because I can anticipate what they want. I can also be more efficient in processing their requests [ . . . ]. Of course, sometimes they [members of the M-banking industry] have asked me to intercede in their behalf. I help when I can, you know? I know some of them, and they are good people.*"

Members of Government Elite

Interestingly, our results demonstrate that creating and nurturing relations with members of the government elite and/or the judiciary were not beneficial to our firm sample. Respondents indicated that informal relations with members of the government, even though effective [82], were not popular, and as such, the interviewed firms chose not to devote many resources towards pursuing them. As a result of the instability at the highest levels of the Guatemalan government, creating and maintaining relations with these actors was not found to be an effective manner to gain stability for firms entering the

new M-banking industry in Guatemala. As Respondent 16 argues, "*It is quite difficult to forge relations when the upper levels of government are a constantly revolving door.*"

Establishing and maintaining informal relations with members of the government elite was not an easy task for most companies to accomplish. This was illustrated by Respondent 26:

> "*Once you are appointed [at the highest levels of the ministry of communications], everyone becomes your 'friend.' You are invited to many social events, and people just follow you everywhere. After those encounters, people request meetings with you, and you do not even remember who they are. I always advise them to follow established channels instead of coming to me, though. They do not understand that I am only one public official and cannot do all they want me to do.*"

Judiciary

Less advanced economies are perceived as "uncertain" due to their weak laws and regulations [29]. Building on the extant literature, it could be assumed that establishing informal relations with members of the judiciary system might help firms operate in these situations. However, the evidence from our research indicates a counter-intuitive finding. All respondents agreed that establishing informal relations with members of the judiciary system did not benefit their operations in Guatemala.

Indeed, such linkages were simply seen as "insurance" or as Respondent 10 put it:

> "*While our company has worked to have a good relationship with several judges and even members of the Supreme Court, these relations are of no use to decrease uncertainty in our operations because there is no law regulating our industry. Instead, we have these connections in case one day we are brought to court [ . . . ] then, we hope to be able to use our leverage in the judiciary system.*"

The members of the judiciary system in Guatemala also confirmed what was expressed by respondents from firms. According to the judges interviewed, firms spend little time or effort creating informal relations with them. The judges attributed this behavior to the small amount of power they have to provide M-banking firms with the advantages they need to operate since there was no specific law regulating this industry. However, according to all respondents, the judiciary system would increase in importance once litigation is involved. As Respondent 32 expressed:

> "*For the most part, businesses leave us alone [ . . . ]. I know several people in the M-banking industry who would like to have a clear law regulating their operations, but they stopped asking me to intervene in this [ . . . ]. Once they are brought to court, things change, however. Then they think they are your friends. Of course, the law is blind, and we do what we can to operate adhering to established laws and protocols.*"

In summary, Figure 1 offers a graphic representation of our results. It shows that to enter the emerging M-banking industry, established firms with embedded practices had to utilize internal and external CE to radically change their traditional strategic mindset. In doing so, these firms developed new entrepreneurial competencies and skills. Given the particular nature of the industry, established firms had to find the most cost-effective manner on which to participate in this new business sector. In this regard, these firms not only uncovered new sources of knowledge and stability but were also able to begin transforming deeply engrained practices in the country in which they operated in. Table 4 shows a summary of the multiple sources of data and illustrations explaining how established firms with previously embedded practices decided to change them.

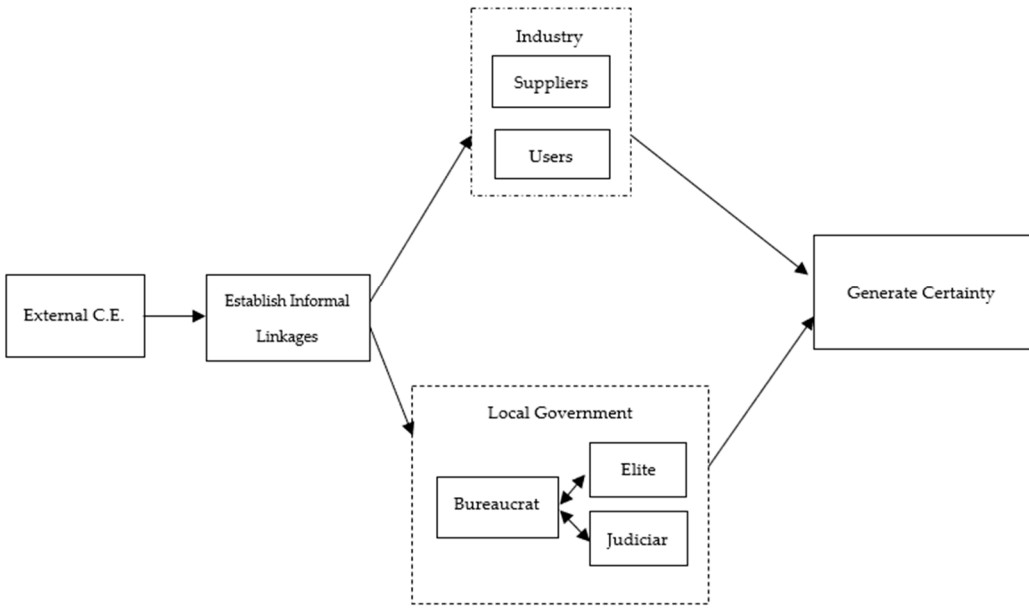

**Figure 1.** Conceptual framework of using informal linkages to operate in a location characterized by industry and institutional uncertainty.

**Table 4.** Data sources with illustrations of how firms changed their embedded practices.

| Dynamics | Sources | Illustrations |
|---|---|---|
| New Sources of Knowledge | Interviews with M-Banking firms Published information | The industry in the country is very unstable . . . To generate the knowledge we need to operate, we rely on information from an important supplier. (Respondent 8) We had a rough start because many of the potential users barely spoke Spanish . . . We had to adapt and invited them to help us create the app in their language . . . I think that has helped us a lot in our penetration with those [indigenous] populations. (Respondent 9) No evidence |
| New Sources of Stability | Interviews with M-Banking firms Interviews with bureaucrats Interviews with government elite Interviews with members of the judiciary Published information | We can no longer count on high level officials to help us . . . they [government elites] are coming and going, and that is a source of uncertainty. We had to forge relations with those who are there when their bosses are gone. They do the bidding for us, and it is more cost effective. (Respondent 17) We are seeing an increased attention to ourselves. Maybe because the ministers and their teams don't last long in their jobs?. (Respondent 35) People in that industry [M-banking] stop asking for favors once they understood I did not have the power to create or change laws. (Respondent 26) . . . I mean, I could have helped those people [M-banking firms], but my job is not to create laws. My job is to make sure the laws are applied correctly, and at this time, we don't have a law for mobile banking. (Respondent 33) No evidence |

## 5. Discussion

The objective of our study was to examine why and how established firms changed their embedded practices to enter a nascent and unregulated industry that was also responsible for reducing financial exclusion among consumers. The examination of these important changes (e.g., increasing financial inclusion to previously discriminated patrons) provides a valuable template to other less advanced economies characterized by a weak formal institutional environment and especially those interested in actively trying to overcome socioeconomic challenges through the development of new entrepreneurial competencies and skills. To the best of our knowledge, our study is the first of this kind. Our results show that addressing inequality and discrimination within organizational boundaries is multi-faceted, and firms need a combination of economic incentives and social pressures to do so even when entering a new industry. Moreover, our results indicate that to succeed in a new industry in a frontier market, such as Guatemala, well-established firms had to utilize a combination of internal and external CE. By doing so, it allowed these firms to rethink outdated and traditional embedded practices, which created new opportunities to leverage uncommon sources of knowledge needed to gain a competitive advantage.

We began our analysis by uncovering why established firms decided to enter a nascent industry in Guatemala. Our analysis demonstrates that the decision was a result of a combination both economic incentives and social pressures. Extant scholarship argues that firms can adapt their practices to take advantage of economic opportunities [75]. Nevertheless, our study shows that although the new industry had economic potential, it was not enough of an incentive for established firms to enter it. Instead, they decided to enter this industry because it not only offered economic potential but also because they felt social pressures from international organizations to do so. This insight is a significant departure from the extant literature, which proposes that changes to organizational strategies are more likely to result because of competitive pressures alone [39]. Thus, in this regard, our results seem to indicate that not only financially stable firms but also those operating in less competitive environments should proactively consider developing new entrepreneurial competences because of the increasing external social pressures from visible and persuasive international organizations to reduce socioeconomic inequalities worldwide (e.g., the UN). The necessity to develop corporate entrepreneurship strategies in frontier markets may further intensify with increasing global competition.

Our study makes an important departure from the existing literature by analyzing how firms with embedded practices utilize internal and external CE when entering a new industry in a frontier market. Extant literature proposes that formal institutions impact a firm's ability to manage the different forms of CE appropriately [12] and can increase uncertainty in locations in which the formal institutional environment is weak [29]. Yet, such analyses have not provided adequate knowledge of how to create change through corporate entrepreneurship in locations with a weak formal institutional environment. We extend theory in this regard by investigating whether firms that have been largely sheltered through national protectionism policies were able to change their embedded practices by adopting cost-efficient approaches. Specifically, we found that these firms choose to utilize internal CE to acquire the necessary technology to enter the emerging industry, which initially increased their costs. At the same time, they utilized external CE to create and leverage informal relations with technological suppliers to gain knowledge about their competitors' actions, which involved no direct costs.

Our study further contributes to the literature by identifying and differentiating how firms with embedded practices utilize corporate entrepreneurial practices to decrease the uncertainty generated by an unstable and formal institutional environment. As previously stated, the extant literature acknowledges the important role that the external environment plays in corporate entrepreneurial strategies [83]. However, most of these studies have focused on industry conditions or particular cultures [12]. Hence, we have limited knowledge about how a country's formal institutions (i.e., laws and regulations) [62] shape corporate entrepreneurial behaviors, especially in less advanced countries where formal institutions

are the most undeveloped. Extant scholarship, in this regard, advocates for establishing relations with relevant local actors [26]. Moreover, the literature recognizes that one of the most relevant actors that entrepreneurial firms should develop relationships with are high-ranking members of the local government [82]. However, due to their unstable tenure, the number of resources needed to nurture relations with these actors, and because of the allegations of corruption in the highest levels of the Guatemalan government [55], all of the firms we studied decided to not pursue such relations. Instead, they utilized external CE to create informal relations with previously overlooked actors as a more effective strategy. Specifically, cultivating relationships with low-level government officials was more advantageous for these firms because they acted as intermediaries to other influential but often temporary and unstable members of the elite and/or judiciary system.

Our results also show that the firms we studied departed significantly from the established standards of operating in Guatemala. Traditionally, similar to other less advanced economies, the notion of implementing business ethics did not extend beyond organization boundaries, and as such, firms operating in this country were dismissive of the demands of their customers [57] and especially those from poorer and indigenous backgrounds. Nevertheless, due to the attractiveness of the new industry and pressures from important international actors, the established firms who chose to enter it had to turn to these once overlooked users to craft attractive, user-focused services. This is an important and significant change from the status quo since these firms are more well known for normalizing and reinforcing socioeconomic inequalities rather than developing programs to alleviate them. Creating practices that offer differential value propositions for marginalized groups, especially customers without economic means and with little education, provides them with opportunities to acquire and build wealth but, up until lately, has largely been of little concern for well-established firms and particularly those operating in the financial service industry.

### 5.1. Practical Implications

For practitioners, this research illustrates that cultivating and maintaining informal relations with preciously overlooked actors, such as customers with little formal education, suppliers, as well as with all members of the bureaucracy of the local government in an unstable location, are beneficial entrepreneurial competencies to develop. Informal relations with customers and suppliers may enhance the competitive advantage of firms in a new industry in an uncertain setting, and those with members of the local government can be particularly advantageous. Informal relations with members of the bureaucracy in this context were the most valuable in successfully carrying out operations in such locations. Moreover, this study highlights the need for firms operating in highly protected locations to re-think their archaic and discriminatory-based business models and realize that previously overlooked stakeholders can be the key to gaining a competitive advantage, especially in nascent industries of international interest.

As for policy development, it is recommended that governments in frontier markets eager to gain emerging-country status among foreign investors encourage the development and enforcement of rules to standardize operations of firms operating in new industries, especially those industries that have the potential to lift people out of poverty. These policies should result in a clear set of rules that decrease uncertainty and encourage greater investment in less advanced economies. In addition, policymakers should welcome assistance from internationally recognized entities that can help firms with embedded practices develop new entrepreneurial competencies, which are necessary to increase the inclusiveness of previously neglected groups nationally and worldwide.

### 5.2. Limitations

Since our research was exploratory in nature, we confined it to one industry and one country at a specific period of time, as proposed by Eisenhardt [46]. The natural setting in which our study took place makes it extremely difficult to control external variables,

which in turn can hinder replication. Future research may address these limitations by testing the reliability of our findings and, in particular, the relevance of the different actors on which firms rely to operate in a challenging institutional environment in other contexts in a longitudinal manner. Additionally, our study only analyzed those firms that entered the M-banking industry in Guatemala. Future research should study firms that were not successful in entering new industries to compare their strategies to those that were able to utilize institutional entrepreneurship to adapt to a new industry.

## 6. Conclusions

Our research aims to contribute to the important body of work that addresses the UN's Sustainable Development Goals. Specifically, our study seeks to advance our understanding of corporate entrepreneurship and how it is an important enabler in fostering greater inclusion of financial resources among disadvantaged and oppressed groups. Through our exploratory analysis, it is our hope that we offer new insights into how to address three key objectives of the UN's SDG aspirations (i.e., SDG1 No Poverty; SDG 8 Decent Work and Economic Growth; and SDG10 Reduced Inequality). More precisely, we present why and how well-established firms operating in a less advanced frontier market in the financial service industry were able to successfully utilize corporate entrepreneurial strategies to change their embedded practices. Our results show that firms operating in a sheltered country need a combination of economic incentives and external pressures to initiate changes to age-old discriminatory practices. Our study also reveals some encouraging findings about firms operating in these challenging environments. We found that when a new and emerging industry is present in a location characterized by a weak formal institutional environment, firms are able to develop and implement new entrepreneurial competencies to become more internationally competitive. In this regard, firms operating in less advanced economies are encouraged to engage in both internal (e.g., establishing informal relations with previously overlooked actors) and external (e.g., leveraging the insights gathered from those actors) corporate entrepreneurship if they aspire to radically alter their existing and often exclusionary embedded practices.

**Author Contributions:** Conceptualization, J.G. and D.R.D.; methodology, J.G.; validation, J.G.; formal analysis, J.G. and D.R.D.; investigation, J.G.; resources, J.G. and D.R.D.; data curation, J.G.; writing—original draft preparation, J.G. and D.R.D.; writing—review and editing, J.G. and D.R.D.; visualization, J.G. and D.R.D.; supervision, J.G. and D.R.D.; project administration, J.G. and D.R.D.; funding acquisition, J.G. and D.R.D. All authors have read and agreed to the published version of the manuscript.

**Funding:** This research received no external funding.

**Institutional Review Board Statement:** Not applicable.

**Informed Consent Statement:** Not applicable.

**Data Availability Statement:** Restrictions apply to the availability of these data. Upon request, specific questions from the interviews are available.

**Conflicts of Interest:** The authors declare no conflict of interest.

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
