# Peer review of "Frontier Markets and Sustainable Entrepreneurial Competences: An Exploratory Study of the Impact of a New Industry in Guatemala"

_sustainability, doi:10.3390/su132011314_

Round 1

Reviewer 1 Report

Please see attach PDF

Author Response

Reviewer's key comments:

I have carefully read the paper “Frontier Markets and Sustainable Entrepreneurial Competences: An Exploratory Study of the Impact of a New Industry in Guatemala”, where different points of view are studied, regarding the occurrence of new industries in Guatemala in the context of frontier markets.

Based on my reading, I think that this paper deals with a topic of scarce academic interest, this is also applied to the professional and the market and companies' management fields. In my opinion this paper is very irrelevant for the scientific community.

Moreover, it uses very outdated information that dates from 2017. Since 2017 to 2021, companies and consumers behaviours and the world order in general have experimented such radical changes that these data have just lost all their viability for any study. I sincerely hope that my comments help.

[Response]  We thank your honest assessment of our paper. In the revised manuscript, we acknowledged that a limitation of our study is that our data analyzes the specific point in time when this new industry emerged. We encourage future researchers to validate our results with a longitudinal approach.

Reviewer 2 Report

General remarks

The paper is structured properly, although the aim is not defined precisely. No hypotheses are proposed.

The idea of the paper is innovative. However, it is not developed properly in the text of the paper.

The logical structure of the paper is coherent.  

Assessment of the paper

The aim of the paper is very broad. It is properly presented in theoretical considerations. The Authors want to capture a too broad scope of the problems illustrated with narrow research.

Detailed remarks:

  1. A good, well-documented, theoretical part. However, it is too broad. In consequence, the causal links can be difficult to prove.
  2. It is obvious that new forms of activities in the frontier markets can be treated as a response to economic and social pressures. It can be related to sustainability by reducing poverty. However, the issues are defined so broadly that we can always find causal links. For example, there is an example of the influence of inspiration from the US Embassy (l. 439-444). Are there any other examples?
  3. While the background of empirical studies – case study is well elaborated, the method of qualitative research raises some doubts. Collecting answers from a group and their analysis is not described too clearly. Perhaps attaching the detailed partial results could be helpful. The explanation of collecting data about the companies, l. 332-343 demands for a further elucidation – perhaps a table?
  4. The terms used in research, e.g. social pressure, laws, regulations, etc. l. 358-361 are so general that they always must be used and they are always connected to other broadly defined ideas. For example, if we study together social pressure and the limitation of socioeconomic inequality, there are always causal links. Much more insightful in such a case would be, for example. M-banking as an instrument of loans availability for small businesses and start-ups.
  5. Even in qualitative analysis, we have to care about clear lines of reasoning and clearly defined causal links. The Authors draw so many far-reaching conclusions from the results of conversations.

The paper can be published considering the following changes:

  1. More precisely described ideas between which the causal links are searched for. In this case visualization with figures and schemes illustrating the theoretical assumptions would be useful
  2. More precisely described process of empirical research – separately attached intermediate data. It is not necessary but perhaps additional table/s would be helpful.
  3. The casual links between the observations and conclusions deriving from the answers and new factors – links between implementation of a new industry in a frontier market and selected features of sustainability, should be presented more precisely. Perhaps some factors, as legal, etc. could be described in more specific terms. Just an example. Showing for instance, instead of general terms, specific regulations concerning state-sponsored programs addressed to the excluded groups for which M-banking could be instrumental? Perhaps a more detailed description of banks' encouragement for exclusion-eliminating investment.

There is a small typing error – county, should be country, l. 207.

Author Response

Reviewer's key comments:          

The paper can be published considering the following changes:

1. More precisely described ideas between which the causal links are searched for. In this case visualization with figures and schemes illustrating the theoretical assumptions would be useful.

2. More precisely described process of empirical research –separately attached intermediate data. It is not necessary but perhaps additional table/s would be helpful.

3. The casual links between the observations and conclusions deriving from the answers and new factors – links between implementation of a new industry in a frontier market and selected features of sustainability, should be presented more precisely. Perhaps some factors, as legal, etc. could be described in more specific terms. Just an example. Showing for instance, instead of general terms, specific regulations concerning state-sponsored programs addressed to the excluded groups for which M-banking could be instrumental? Perhaps a more detailed description of banks' encouragement for exclusion-eliminating investment.

4. There is a small typing error – county, should be country, l. 207.

[Response] We are grateful for your helpful comments. As you have proposed, we now include Figure 1 in our revised manuscript. In this Figure, we aim to illustrate the theoretical assumptions and the results obtained from our study. We appreciate your important comment that we should demonstrate causal links between observations and conclusions. Unfortunately, due to the qualitative nature of our research design, we were unable to resolve this concern fully. Our aim, however, with the type of approach that we employed in this study was to generate theoretical insights as opposed to showcasing the casual links between factors. We encourage future researchers, in our revised manuscript, to validate our findings with a longitudinal approach. Finally, we thank you for pointing out our typo (county instead of country). We fixed it promptly.

Reviewer 3 Report

My comments on the paper - Frontier Markets and Sustainable Entrepreneurial Competences: An Exploratory Study of the Impact of a New Industry in Guatemala- are as follows.

The paper presents an interesting analysis and we consider that the research is of interest. However, we recommend some revisions.

The abstract is clear and presents the purpose of the paper.

The introduction provides the necessary background information but the introduction does not explicitly mention the objective of the research. The author indicate the added value that the paper brings to the existing academic literature.

In the introduction section, the structure of the paper on sections is not provided.

The structure of the paper is coherent and the research methodology used by the author is adequate for the approached subject.

The discussions and the conclusions are significant but it is not pointed out if the results of the research are in accordance or not with other studies. We recommend that the results obtained from the study should be compared with the results obtained in the case of similar researches from the academic literature.

We recommend reviewing paper format, in accordance to the instructions for authors (especially the list of references).

Author Response

Reviewer’s key comments:

In the introduction section, the structure of the paper on sections is not provided.

The structure of the paper is coherent and the research methodology used by the author is adequate for the approached subject. The discussions and the conclusions are significant but it is not pointed out if the results of the research are in accordance or not with other studies. We recommend that the results obtained from the study should be compared with the results obtained in the case of similar researches from the academic literature. We recommend reviewing paper format, in accordance to the instructions for authors (especially the list of references).

[Response} We thank you for your encouraging and constructive comments. Based on the reviews requested, we added more structure to the introduction. We also noted that studies similar to ours, particularly those originating from frontier markets are still in their infancy. We also copyedited our revised manuscript. 

Round 2

Reviewer 1 Report

Paper has not improved